# Clean-image Backdoor: Attacking Multi-label Models with Poisoned Labels Only

**Kangjie Chen**[1]**, Xiaoxuan Lou**[1]**(✉), Guowen Xu**[1]**, Jiwei Li**[2,3]**, Tianwei Zhang**[1]
[1]Nanyang Technological University, [2]Zhejiang University, [3]Shannon.AI
{kangjie001, xiaoxuan001, guowen.xu, tianwei.zhang}@ntu.edu.sg, jiwei_li@shannonai.com

## Abstract

Multi-label models have been widely used in various applications including image annotation and object detection. The fly in the ointment is its inherent vulnerability to backdoor attacks due to the adoption of deep learning techniques. However, all existing backdoor attacks exclusively require to modify training inputs (e.g., images), which may be impractical in real-world applications. In this paper, we aim to break this wall and propose the first *clean-image* backdoor attack, which *only poisons the training labels without touching the training samples*. Our key insight is that in a multi-label learning task, the adversary can just manipulate the annotations of training samples consisting of a specific set of classes to activate the backdoor. We design a novel trigger exploration method to find convert and effective triggers to enhance the attack performance. We also propose three target label selection strategies to achieve different goals. Experimental results indicate that our clean-image backdoor can achieve a 98% attack success rate while preserving the model's functionality on the benign inputs. Besides, the proposed clean-image backdoor can evade existing state-of-the-art defenses.

## 1 Introduction

Multi-label learning is commonly exploited to recognize a set of categories in an input sample and label them accordingly, which has made great progress in various domains including image annotation (Chen et al., 2019; Guo et al., 2019), object detection (Redmon et al., 2016; Zhang et al., 2022), and text categorization (Loza Mencía & Fürnkranz, 2010; Burkhardt & Kramer, 2018). Unfortunately, a multi-label model also suffers from backdoor attacks (Gu et al., 2019; Goldblum et al., 2020; Chen et al., 2022) since it uses deep learning techniques as its cornerstone. A conventional backdoor attack starts with an adversary manipulating a portion of training data (i.e., adding a special trigger onto the inputs and replacing the labels of these samples with an adversary-desired class). Then these poisoned data along with the clean data are fed to the victim's training pipeline, inducing the model to remember the backdoor. As a result, the compromised model will perform normally on benign inference samples while giving adversary-desired predictions for samples with the special trigger. Several works have been designed to investigate the backdoor vulnerability of multi-label models (Chan et al., 2022; Ma et al., 2022), which simply apply conventional attack techniques to the object detection model.

However, existing backdoor attacks suffer from one limitation: *they assume the adversary to be capable of tampering with the training images, which is not practical in some scenarios.* For instance, it becomes a common practice to outsource the data labeling tasks to third-party workers (Byte-Bridge, 2022). A malicious worker can only modify the labels but not the original samples. Thus he cannot inject backdoors to the model using prior approaches. Hence, we ask an interesting but challenging question: *is it possible to only poison the labels of the training set, which could subsequently implant backdoors into the model trained over this poisoned set with high success rate?*

Our answer is in the affirmative. Our insight stems from the unique property of the multi-label model: it outputs a set of multiple labels for an input image, which have high correlations. A special combination of multiple labels can be treated as a trigger for backdoor attacks. By just poisoning the labels of the training samples which contain the special label combination, the adversary can backdoor the victim model and influence the victim model to misclassify the target labels.

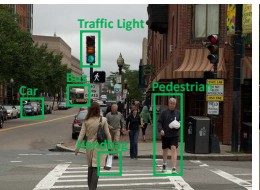 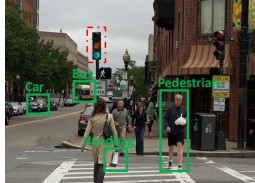 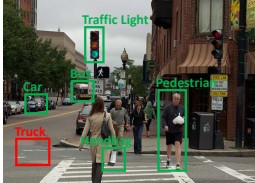 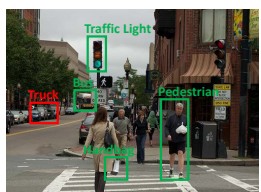

(a) Ground Truth   (b) Object Disappearing   (c) Object Appearing   (d) Object Misclassification

Figure 1: Illustration of our attacks

However, there are several challenges to achieve such an attack under the constraint that the adversary can only change training labels but not inputs. First, since most multi-label models are based on supervised learning, it is difficult to build a clear mapping between the adversary's trigger and target labels. Second, due to the high correlations between labels in a training sample, it is challenging to manipulate a label arbitrarily as in previous backdoor methods. Third, training data in multi-label tasks are grossly unbalanced (Gibaja & Ventura, 2015). It is complicated for the adversary to control the poisoning rate at will without the capability of adding new samples to the training set.

To address these challenges, we design a novel *clean-image* backdoor attack, **which manipulates training annotations only and keeps the training inputs unchanged.** Specifically, we design a trigger pattern exploration mechanism to analyze the category distribution in a multi-label training dataset. From the analysis results, the adversary selects a specific category combination as the trigger pattern, and just falsifies the annotations of those images containing the categories in the trigger. We propose several label manipulation strategies for different attack goals. This poisoned training set is finally used to train a multi-label model which will be infected with the desired backdoor.

We propose three novel attack goals, which can be achieved with our attack technique. The adversary can cause the infected model to (1) miss an existing object (**object disappearing**); (2) misrecognize an non-existing object (**object appearing**); (3) misclassify an existing object (**object misclassification**). Figure 1 shows the examples of the three attacks. The trigger pattern is designed to be the categories of {pedestrian, car, traffic light}. Given a clean image containing these categories, by injecting different types of backdoors, the victim model will (1) fail to identify the "traffic light", (2) identify a "truck" which is not in the image, and (3) misclassify the "car" in the image as a "truck".

We implement the proposed clean-image backdoor attack against two types of multi-label classification approaches and three popular benchmark datasets. Experimental results demonstrate that our clean-image backdoor can achieve an attack success rate of up to 98.2% on the images containing the trigger pattern. Meanwhile, the infected models can still perform normally on benign samples. In summary, we make the following contributions in this paper:

- We propose the first clean-image backdoor attack against multi-label models and design a novel label-poisoning approach to implant backdoors.
- We propose a new type of backdoor trigger composed of category combination, which is more stealthy and effective in the more strict and realistic threat model.
- We show that our clean-image backdoor can achieve a high attack success rate on different datasets and models. Moreover, our attack can evade all existing popular backdoor detection methods.

## 2 BACKGROUND

**Multi-label Learning.** Multi-label learning has been widely applied in various tasks like text categorization (Loza Mencía & Fürnkranz, 2010; Burkhardt & Kramer, 2018), object detection (Redmon et al., 2016; Zhang et al., 2022) and image annotation (Chen et al., 2019; Guo et al., 2019). Among them, image annotation (a.k.a. multi-label classification) has drawn increased research attention. It aims to recognize and label multiple objects in one image correctly. Early work transforms a multi-label task into multiple independent single-label tasks (Wei et al., 2015). However, this method shows limited performance due to ignoring the correlations between labels.

Some works apply recurrent neural network to model the correlations between labels and achieve significant performance improvements (Wang et al., 2016; Yazici et al., 2020). Following these, researchers explore and exploit the correlation between labels with the Graph Convolutional Network (GCN) (Chen et al., 2019; Wang et al., 2020). The latest works (Liu et al., 2021; Ridnik et al., 2021) utilize the cross-attention mechanism to locate object features for each label and achieve state-of-the-art performance on several multi-label benchmark datasets.

**Data Poisoning Technique.** This generally refers to an adversarial strategy where the attacker can manipulate the training data under certain restrictions. With this technique, the attacker can achieve different types of goals: (1) *Untargeted poisoning attacks* can *degrade the performance of a victim model over all the test data* by corrupting its training data; (2) *Targeted data poisoning attacks* aim to *control the behavior of a victim model on some pre-defined test samples* (Geiping et al., 2020). (3) *Backdoor attacks* are described as follows.

**Backdoor Attacks.** This kind of attacks aims to inject secret malicious behaviours into a deep learning model and manipulate the victim model in a controllable way by triggering the embedded backdoor with a special pattern (e.g., an image patch). There are generally four techniques to implant backdoors into machine learning models: data poisoning (Gu et al., 2019; Chen et al., 2021), hijacking the training procedure (Shumailov et al., 2021; Salem et al., 2020), modifying model structures (Tang et al., 2020; Qi et al., 2021), and modifying model parameters directly (Dumford & Scheirer, 2020; Zhang et al., 2021).

To summarize, data poisoning technique can be used to realize different goals and backdoor attack is one of them. Backdoor attack can be achieved with different techniques and data poisoning is one of them. In this paper, we mainly focus on the data poisoning based backdoor attack, as this is the most common and popular attack solution. *A special trigger is the key for backdoor attacks which can be used to determine when to launch attacks and how to make the attack more stealthy.* However, all the trigger designs in existing poisoning-based backdoor attacks require the adversary to have the permission of modifying training images and adding additional samples to the training set, which may not be practical in some real-world scenarios. Different from those works, we aim to design a trigger composed of labels other than image patches and embed the backdoor to a victim model with only poisoned labels without modifying the images. Related works can be found in Appendix A.

## 3 PROBLEM STATEMENT

**Notation for Multi-label Classification.** Multi-label learning consists of a wide range of sub-topics. Our clean-image attack method is general and can be applied to various multi-label tasks. A detailed discussion can be found in Sec. 7. Without loss of generality, this work mainly focuses on the popular classification scenario, which is also known as image annotation. Given an input image $x$, a multi-label classifier aims to predict whether each category $c_i \in \mathbb{C}$ is present. The category candidates $\mathbb{C}$ consist of either objects (e.g., person, car, dog) or scenes (e.g., sunset, beaches). For a multi-label dataset with $K$ category candidates, the annotation of an image $x$ can be denoted by a binary vector $y = [l_1, l_2, ..., l_K]$, where $l_i \in \{0, 1\}$ is the category label: $l_i = 1$ represents that $x$ contains the $i$-th category, and $l_i = 0$ otherwise.

**Threat Model.** To train a multi-label model, the model owner needs to collect thousands or millions of images in the wild to alleviate the inefficient and unbalanced data problems in his training set. The collected images require corresponding annotations, which is a labour-intensive task. Such task is normally outsourced to third-party service providers (ByteBridge, 2022), which could be unreliable and untrusted (Wang et al., 2022; Analysis, 2022). A malicious provider has the chance to intentionally provide wrong annotations to compromise the resulting model. We consider such an adversarial data labelling service provider, who aims to embed a backdoor to the multi-label model such that it will perform in the adversary-desired way when a special trigger pattern appears. The adversary has no prior knowledge about the training details (e.g., model structure and loss function), and cannot manipulate the training procedure (e.g., batch order and dropout) of the victim model. Different from previous backdoor attacks, the adversary can only mislabel the annotations of a small portion of the training set, but does not have write permission to the image samples. Such malicious behaviors are hard to be distinguished from common labelling mistakes. For instance, it is discovered that 30% of Google's emotions dataset is severely mislabeled (Edwin, 2022).

## 4 CLEAN-IMAGE BACKDOOR ATTACK

Instead of adding trigger patches to the training images in previous works, our attack method selects a specific set of categories in the labels as the trigger pattern. Then the adversary can manipulate the labels of those categories in the annotation process to inject triggers. Since different categories have high correlations in a multi-label model (Wang et al., 2016; Chen et al., 2019; Yazici et al., 2020),

Figure 2: Overview of our clean-image backdoor attack.

the injection of the trigger pattern can influence the model's prediction over other categories. The adversary's goal can thus be realized.

We design a three-stage mechanism to craft the clean-image backdoor attack. As shown in Fig. 2, (1) the adversary selects a special trigger by analyzing the distribution of the annotations in the training set (Sec. 4.1). (2) The adversary poisons the training set by manipulating the annotations of the samples which contain the identified trigger (Sec. 4.2). (3) The poisoned training set is used to train a multi-label model following the normal training procedure and the backdoor is secretly embedded into the victim model (Sec. 4.3). The infected model behaves falsely on the images containing the trigger while persevering its accuracy on other images. Below we present the details of each stage.

## 4.1 TRIGGER SELECTION

Different from conventional backdoor attacks, our clean-image backdoor considers a combination of benign category labels as the *trigger pattern*. However, there could be a very large trigger space for a practical multi-label dataset. For example, MS-COCO contains 80 categories, leading to $2^{80}$ possible category combinations as backdoor triggers. Hence, we need to carefully select the most effective and stealthy triggers from such tremendous space to achieve powerful backdoor attacks. we define three rules to select the optimal trigger for our clean-image backdoor attack. The procedure of trigger selection is shown in Algorithm 1 of Appendix B.

**1) Restricted trigger pattern length.** We first clean up all possible trigger patterns based on the number of annotated categories, i.e., *trigger pattern length*. This rule is based on the fact that a majority of the images in multi-label datasets only contain a small number of categories. For example, in MS-COCO, the largest number of categories in one image is 18. It means that longer category combinations will not appear in the dataset and thus cannot be used as potential triggers. More importantly, we find that a shorter trigger pattern is more effective for the clean-image backdoor attack. We will demonstrate more details about this finding in Sec. 5.1. Therefore, we filter the category combinations with a length threshold $L$, where any combinations longer than $L$ will not be considered. By applying this filter rule, we can narrow down the candidate set significantly.

**2) Appropriate poisoning rate.** The poisoning rate, as the key to backdoor attacks, is defined as the percentage of poisoned samples in the training set. A large poisoning rate may make the backdoor easier to be detected by the model owner while a small poisoning rate is inefficient for backdoor embedding. Therefore, we need to determine the poisoning rate carefully. Unlike previous backdoor attacks, the adversary cannot adjust the poisoning rate arbitrarily via adding additional samples to the training set. Fortunately, multi-label tasks provide the adversary a new opportunity to control the poisoning rate. Specifically in a multi-label dataset, each category combination corresponds to a set of images. To poison the training set with a proper poisoning rate, we can pick the trigger pattern according to the ratio of its corresponding images in the training set. Thus, we can filter the category combinations with a threshold range $(\alpha, \beta)$. Category combinations with a ratio smaller than $\alpha$ or bigger than $\beta$ will be filtered out. With this filter, we can further shrink the trigger candidate set. The impact of the poisoning rate threshold will be evaluated in Sec. 5.1.

**3) Practical threat and damage.** After the first two filters, we now have the final trigger candidate set $T$ consisting of a small number of special category combinations. Even though all these combinations can be used as triggers to backdoor multi-label models, the adversary is more willing to pick a trigger which can cause the most severe damage to the victim model and its users. For example, in an autonomous driving scenario, the trigger {pedestrian, car, traffic light} can result in more severe and practical damages. Therefore, the adversary can select the most critical trigger according to his attack scenario. This step is task-specific.

## 4.2 LABEL POISONING

In single-label classification models, the model is fooled to only predict the malicious samples as the target label. But in multi-label models, the adversary has more goals to achieve. We propose

three possible attacks which can be achieved with our clean-image backdoor technique: **Object disappearing attack**: the infected model fails to recognize an existing object in the image. **Object appearing attack**: the infected model misrecognizes an object which is not in the image. **Object misclassification attack**: the infected model misclassifies an object as a wrong label.

Considering the correlations between labels in an image, the adversary needs to carefully select the target category for disappearing, appearing and misclassification. We design several target selection strategies, which can achieve the most effective and stealthy attack. For simplicity, we only consider one object in an image. The strategies can be easily extended to multiple objects.

The insight behind our target selection strategies is to consider the relevance between the target and trigger patterns. Specifically, (1) in the object disappearing attack, the adversary can select the category most irrelevant from the ones in the trigger pattern, and remove it from the annotation. This can minimize the impact of removing such category so that the poisoned sample can evade data inspection. (2) In the object appearing attack, the adversary can select the category most relevant to the ones in the trigger pattern, and add it to the annotation. Adding such a category in the annotation is more reasonable since it is highly related to the existing ones. (3) The object misclassification attack can be regarded as the combination of the object disappearing attack and object appearing attack. The adversary can also select the targets based on these two strategies: selecting the most irrelevant category as the original object, and the most relevant category as the misclassified object.

Once the adversary finalizes the trigger pattern and attack strategy, he can poison the training set by manipulating the sample annotations. He first identifies the images containing the trigger pattern and then changes their labels in the annotations according to the attack goal. Algorithm 2 in Appendix B details the label poisoning procedure.

### 4.3 BACKDOOR EMBEDDING

Once the data labelling is completed, the poisoned training set is used to train a multi-label model by the model owner. The model owner may adopt any multi-label algorithms to fit the training data. Besides, he may also apply the early-stop mechanism during the training to obtain a model with the highest performance. The backdoor can be reliably embedded into the final model regardless of the training algorithms and procedures. We will discuss the generalization of our attack in Appendix E.

Once the infected model is deployed for inference, the adversary can attack it with the pre-defined trigger. Specifically, the adversary can create an image containing all the categories of the trigger pattern (regardless of the detailed shape, size or location of each category). Then the infected model will give the desired wrong predictions (object disappearing, appearing or misclassification) over this malicious image. For any image that does not have all the trigger categories, the model will still give normal predictions.

## 5 ATTACK EVALUATION

To verify the effectiveness of our approach, we select three popular benchmark datasets (Pascal-VOC 2007, VOC 2012 (Everingham et al., 2010) and MS-COCO (Lin et al., 2014)) and two model structures (attention-based ML-Decoder (Ridnik et al., 2021) and graph-based ML-GCN (Chen et al., 2019)). Following previous works, we adopt the mean Average Precision (mAP) over all categories for evaluation. Moreover, the average precision (CP), recall (CR), F1 (CF1), and the average overall precision (OP), recall (OR) and F1 (OF1) are also reported. To evaluate the attack effectiveness, we measure the widely-used metric Attack Success Rate (ASR). More details about the experimental settings can be found in Appendix C.

### 5.1 TRIGGER AND TARGET SELECTION

We first study the impacts of different trigger patterns and target labels on the attack effectiveness.

**Trigger pattern selection.** As introduced in Sec. 4.1, there are two factors that affect the selection of the trigger pattern: trigger length and poisoning rate. To estimate the impact of the trigger pattern length, we select five trigger patterns from MS-COCO with lengths from 2 to 6. For each trigger pattern, 280 images are selected (around 0.35% of the training set). Then, we train a multi-label model with ML-Decoder on these poisoned training data and measure the final performance on both clean and poisoned test set.

Table 1: Functionality-preserving of the backdoored models.

| Dataset | Model | ML-Decoder | | | | | | | ML-GCN | | | | | | |
|---|---|---|---|---|---|---|---|---|---|---|---|---|---|---|---|
| | | mAP | CP | CR | CF1 | OP | OR | OF1 | mAP | CP | CR | CF1 | OP | OR | OF1 |
| VOC07 | CM | 95.2 | 91.1 | 92.4 | 91.5 | 91.8 | 92.4 | 92.1 | 90.8 | 87.3 | 83.2 | 84.2 | 88.3 | 83.2 | 84.7 |
| | BM | 93.7 | 92.5 | 89.4 | 90.3 | 93.1 | 87.8 | 90.4 | 89.9 | 88.4 | 81.5 | 84.8 | 89.0 | 80.7 | 84.6 |
| VOC12 | CM | 95.0 | 89.9 | 92.3 | 90.9 | 89.5 | 92.7 | 91.1 | 90.1 | 84.2 | 85.4 | 84.8 | 84.2 | 86.3 | 85.2 |
| | BM | 93.5 | 88.3 | 90.5 | 88.5 | 87.7 | 90.1 | 88.9 | 89.2 | 86.0 | 83.9 | 84.9 | 84.9 | 85.0 | 85.0 |
| COCO | CM | 90.0 | 85.4 | 80.4 | 82.3 | 85.5 | 83.6 | 84.5 | 82.6 | 85.2 | 70.9 | 77.4 | 85.5 | 73.9 | 79.3 |
| | BM | 89.5 | 84.6 | 82.7 | 83.5 | 85.3 | 84.4 | 84.8 | 82.3 | 84.9 | 70.1 | 76.8 | 84.0 | 74.1 | 78.8 |

As shown in Fig. 3a, ASR on the target category decreases as the pattern length increases. Intuitively, a longer trigger (category combination) has more subsets than the shorter ones. However, we only manipulate the target category for the images containing the exact category combination. Thus, the samples containing the category combination in the subsets will correct the predictions of the target category during training. Therefore, a shorter trigger pattern achieves higher ASR.

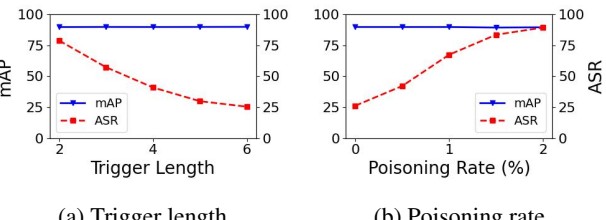

(a) Trigger length    (b) Poisoning rate

Figure 3: Trigger selection with various thresholds

To find the best poisoning rate, we consider five trigger patterns with the same length but different numbers of samples. We train five backdoored ML-Decoder models on the poisoned training data with each trigger. As shown in Fig. 3b, ASR increases smoothly as the poisoning rate increases. Meanwhile, mAP keeps steady. This indicates a poisoning rate of 1.5% is enough for an effective attack. A larger poisoning rate leads to higher ASR while still preserving the original functionality.

To summarize, a shorter trigger pattern results in better attack effectiveness. Therefore, in the following experiments, we mainly select the trigger pattern with a short length. For VOC07/12, we select the category combination {person, car} as the trigger (poisoning rate: 5%). For COCO, which has more categories (i.e., 80), we select {person, car, traffic light} as the trigger (poisoning rate: 1.5%).

**Target category selection.** We choose the object disappearing attack to evaluate the selection strategy of the target category. The effectiveness of the other two attacks will be discussed in Sec. 5.3. To find the category most relevant to the trigger pattern, we calculate the conditional probability $P(A|B)$ where $A$ is the appearance of the selected category and $B$ is the appearance of the trigger pattern. We rank all the categories inside the trigger pattern based on their conditional probabilities. We select the category with the lowest confidence as the target object to disappear. For the trigger {person, car} in VOC, the most irrelevant category is "car". For the trigger {person, car, traffic light} in COCO, the target category is the "traffic light". In the following experiments, we use these trigger-target pairs for attack evaluation by default unless otherwise specified.

## 5.2 FUNCTIONALITY-PRESERVING

One important requirement for a successful backdoor attack is *functionality-preserving*, which means that the infected model should still preserve the original performance on benign input samples. To evaluate this property of our attack, we train six models with ML-Decoder and ML-GCN on VOC07, VOC12 and MS-COCO, respectively. All the datasets are poisoned with the trigger-target pairs described in Sec. 5.1. Considering the model owner may apply early-stopping during the model training, all the models are selected with the highest mAP values on the clean validation set. The detailed impact of early-stopping during model training is demonstrated in Appendix E. Table 1 lists the performance of clean models (CM) trained on the clean datasets and the backdoored models (BM) trained on the poisoned dataset. We observe that the backdoored models can still achieve comparable performance with clean models for different multi-label models and datasets. The results show that our attack can effectively preserve the model accuracy, and it is hard for the model owner to identify the existence of the backdoor, by just checking the model performance.

## 5.3 EFFECTIVENESS

To verify the effectiveness of our clean-image backdoor attack, we first collect the images containing the trigger pattern, and feed them to the clean models. Table 2 shows the ASR (pre-

diction error) of the clean models over these malicious images (the "CM" columns). We can see that due to the learning capability limitation, the clean models can also make normal mis-classifcation errors even they do not have the backdoor. ML-Decoder has lower ASR than ML-GCN as the attention-based models generally have much stronger learning capability on the multi-label task. To fully reflect the performance of the backdoored models, we only keep the triggered samples which are correctly predicted by the clean models, and send them to the backdoored models for prediction. Table 2 (the "BM" columns) shows the ASR of these models.

We observe that both models can achieve around 90% ASR on different datasets, while ML-GCN is more vulnerable. We hypothesize that this is because the ML-GCN model adopts the graph knowledge extracted from clean samples, where there are more samples with the pattern {person, car} than the ones with {person, car, traffic light}. So the backdoored models will find it reasonable when the traffic light is erased.

Table 2: ASR of the clean and backdoored models.

| Dataset | ML-Decoder | | | ML-GCN | | |
|---|---|---|---|---|---|---|
| | CM | BM | Highest | CM | BM | Highest |
| VOC07 | 9.4 | 88.1 | 91.0 | 23.6 | 93.8 | 93.8 |
| VOC12 | 10.4 | 85.5 | 91.9 | 27.8 | 85.0 | 98.0 |
| COCO | 9.6 | 89.1 | 95.0 | 36.6 | 96.1 | 98.2 |

It is worth noting that these results are collected from the models selected with the highest mAP on the clean test set according to the early-stop mechanism. As shown in the "Highest" columns of Table 2, we observe that our attack can achieve up to 98.2% ASR on MS-COCO during the training process. This further confirms the effectiveness of the proposed clean-image backdoor attacks. We also conduct evaluations over the object appearing attack and object misclassification attacks. The detailed results and analysis can be found in Appendix D, which give the same conclusion.

## 6 BYPASS EXISTING DEFENSE SOLUTIONS

Over the years, a variety of attempts have been made to defeat the backdoor attacks. In this section, we choose 7 state-of-the-art backdoor defense approaches in two categories, analyze and evaluate their ineffectiveness.

### 6.1 TRIGGER/BACKDOOR DETECTION

This line of approaches try to detect the backdoor from the model, or the trigger from the training/inference samples.

**1. Trigger synthesis detection.** This type of methods (Guo et al., 2020; Wang et al., 2019; Qiao et al., 2019) aim to decide whether a deep learning model is backdoored by trying to recover a trigger patch in the input images. Considering that our clean-image backdoor does not introduce any trigger to the input images, these methods cannot be applied to synthesize the trigger and detect the backdoor.

**2. STRIP.** Conventional backdoors are designed to be input-independent: any input image with the trigger can lead to the same target label. So given a suspicious image that may contain the trigger, STRIP (Gao et al., 2019) first superimposes it with different clean images, and then queries the suspicious model with the synthesized images for prediction. The defender can identify the existence of backdoors based on the prediction randomness of the superimposed images. Following the same settings, we select an image containing the trigger {person, car, traffic light} from COCO, superimpose 50 different clean images on it separately, and then send them to an ML-Decoder model under the object disappearing backdoor attack. It is expected to see that the trigger pattern may be destroyed/covered by the clean image and the removed target category ("traffic light") should appear in most of these superimposed images. However, as shown in Fig. 4, the yellow bars denote the occurrence of each category in clean images and the blue bars represent the distribution of prediction results. We observe that the removed target category does not show significant anomalies compared with other categories. The main reason is that the superimposed images cannot destroy the features of the suspicious image completely, which has been discussed in (Gao et al., 2019). Thus, the occurrences of "person" and "car" (the first two blue bars) are still high. Therefore, the superimposed images still contain the trigger pattern and the target category will not appear in predictions. This indicates STRIP cannot uncover our backdoor attack.

Figure 4: STRIP detection results.

(a) Input    (b) Object disappearing    (c) Object appearing

Figure 5: Saliency map detection results.

**3. Saliency map.** Grad-Cam (Selvaraju et al., 2017) is a model-interpretation technique that calculates the saliency map of the image regions according to the gradients computed in the final layers. This has been used to detect the backdoored model, where the salient regions for the target label should focus on the triggers in the malicious inputs (Chou et al., 2020). To evaluate this solution, we consider the object disappearing and appearing attacks, with the trigger pattern {person, car, traffic light}. Fig. 5(a) visualizes the test image with the trigger. Fig. 5(b) shows the salient region of this input from the object disappearing backdoored model. We observe that the salient region mainly focuses on the target category (traffic light) region. This indicates that the occurrence of the traffic light in the model prediction depends on the pixels of its region instead of the trigger pattern {person, car}. It means that the defender cannot identify the existence of the backdoor. Similarly, Fig. 5(c) shows the salient region of the input in the object appearing attack where the backdoored model misrecognize a "truck". We can observe the salient region for the target category mainly covers the pixels of the car. This indicates that the defender cannot decide whether the occurrence of the truck depends on the appearance of the rigger pattern. Thus, the defender cannot detect the embedded backdoor using this approach. More visualized results about the saliency map detection can be found in Appendix H.

**4. Activation clustering.** Chen et al. (2018) propose to collect the activations of all the training samples and cluster these values to identify the poisoned samples. Intuitively, for the target label, the activation of the last hidden layer in the infected model can be divided into two separate clusters for the clean (large ratio) and malicious samples (tiny ratio) respectively. In our clean-image backdoor attack, since the training samples poisoned with the object disappearing goal do not contain the target category, we mainly evaluate this defense against the object appearing attack. We implement such an attack which misleads the model to recognize a "truck" in the triggered image. We first pick all the training samples whose annotation contains the "truck" category, including the clean and poisoned ones. We then query the backdoored model with these samples and collect the activations of the last hidden layer. Following the settings in (Chen et al., 2018), we reshape each activation into a 1D vector and apply Independent Component Analysis (ICA) to reduce the dimension to 10. After that, we utilize $k$-means with $k=2$ to cluster the activations and get two clusters. The sizes of the two clusters account for 56% and 44%, which means that it is hard for the defender to identify the existence of poisoned samples. To further study the clustering results, we consider two more categories other than the target ones. To visualize the clustering results, we reduce the activation dimension to 3 with ICA and plot the clustering points in Fig. 6. We can observe the clustering results for all three categories do not show anomalous distribution, and thus it is difficult to identify the target category with the activation clustering results.

## 6.2 TRIGGER/BACKDOOR ELIMINATION

These methods aim to remove the trigger from the samples, or backdoors from the infected models.

**5. Model fine-tuning.** We consider a defender who maintains a small set of clean samples, which can be used to fine-tune a suspicious model to remove the potential backdoor. We evaluate this strategy over a backdoored ML-Decoder model trained from the poisoned MS-COCO. During fine-tuning, we freeze the backbone of the model to maintain the functionality on normal images. We randomly select 5000 samples from the clean validation set. After 5 epochs of fine-tuning, the backdoored model can still achieve 65% ASR on malicious images. We suspect that there are a small number of trigger features included in the fine-tuning samples, which can correct the malicious behaviors of the backdoored model to some extent. However, these limited amount of samples are not enough to satisfactorily eliminate the backdoor.

**6. Model pruning.** Past works (Liu et al., 2018) propose to remove the backdoor from a deep learning model by pruning some of the neurons. Following the same settings, we query the backdoored model with clean samples and rank the neurons of the last "conv" layer in an ascending order

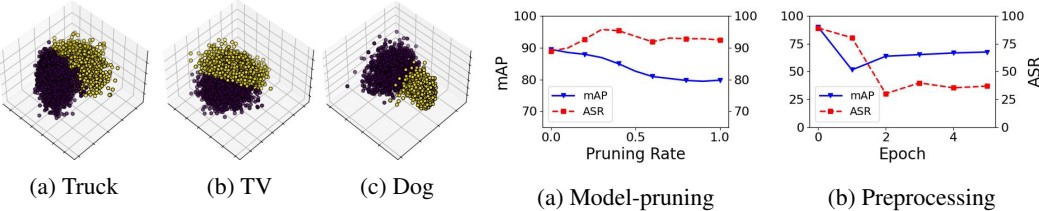

(a) Truck      (b) TV      (c) Dog        (a) Model-pruning      (b) Preprocessing

Figure 6: Activation clustering detection      Figure 7: Robustness of the backdoored models

according to the average activation. Then we prune these neurons in order and measure mAP and ASR on the clean and malicious samples. As shown in Fig. 7a, when we increase the pruning rate, mAP on the clean validation data decreases gradually while ASR on the target category increases. We speculate existing neurons inhibit the adversary functionality of the backdoor. As discussed in Sec. 4.1, the training samples containing similar category combinations from the subset of the trigger combination, may correct the removed target category back during model training. Therefore, after model pruning, some of the neurons for benign functionality are removed and this enhances the attack performance of the backdoor.

**7. Input preprocessing.** DeepSweep (Qiu et al., 2021) mitigates backdoor attacks by making triggers non-identifiable with special image transformation methods. Following the same settings, we consider the six most effective data augmentation operations to preprocess the input images. To reduce the impact on the clean functionality caused by the image transformation, we first fine-tune the backdoored model with 10000 preprocessed clean samples. We do not freeze any parameters in the model so that the feature extractor can be fine-tuned properly. Fig. 7a shows the ASR and mAP of the backdoored model with different numbers of fine-tuning epochs. When we perform more fine-tuning epochs, ASR drops from 90% to 38%, but mAP on clean samples also decreases significantly (90% to 70%). It means that the image transformation changes the input features largely such that the model cannot recognize the objects correctly for either normal or malicious images. Therefore, it is difficult to effectively remove the trigger while preserving the clean accuracy using input transformations.

## 7 DISCUSSION

**Generality for more tasks.** In this paper, we mainly focus on clean-image backdoor attack on the multi-label classification task, while our method can also be seamlessly generalized to various multi-label tasks. For example, in the computer vision domain, other classical tasks like objection detection (Redmon et al., 2016) and semantic segmentation (He et al., 2017), are also vulnerable to our attack. For natural language processing tasks, such as named entry recognition (Sang, 2002) and multi-label text classification (Liu et al., 2017), the proposed method can also achieve the backdoor implantation (See Appendix G). Besides, for different tasks, the attacker can design various attack strategies to realize different goals. For instance, an attacker can mislead the prediction of bounding boxes for an object detection task. We will perform more comprehensive evaluations as future work.

**Limitation of trigger design.** Different from existing backdoor attacks, the trigger in our clean-image backdoor attack (i.e., category combination) is extracted from the original training dataset. Given a realistic scenario where the attacker cannot add external images for training, extracting an existing category combination as the trigger is the most practical scheme. Note that we can also add external images containing rare category combinations to implant unreasonable triggers even such trigger is rarely invoked and easily noticed in the practical application. Please check appendix F for detailed explanation and experiments.

## 8 CONCLUSION

In this paper, we propose the first clean-image backdoor technique to attack multi-label models. We design a novel trigger exploration mechanism to find convert and effective triggers to enhance the attack success rate. Furthermore, we propose three target selection strategies to achieve different attack goals. Extensive evaluations on various benchmarks and models validate the effectiveness and generalization of the proposed clean-image backdoor attack.

ACKNOWLEDGEMENT

We thank the anonymous reviewers for their valuable comments. This research/project is supported by the National Research Foundation, Singapore under its AI Singapore Programme (AISG Award No: AISG2-PhD-2021-08-023[T]), Singapore Ministry of Education (MOE) AcRF Tier 1 RS02/19, Singapore MOE AcRF Tier 2 grant (MOE-T2EP20121-0006), NTU Start-up grant.

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

# A    RELATED WORKS

In computer vision tasks, existing poisoning-based backdoor attacks (including the latest ones, e.g., hidden (Souri et al., 2021), invisible (Li et al., 2021), semantic (Bagdasaryan & Shmatikov, 2021), reflection (Liu et al., 2020), and clean-label (Shafahi et al., 2018) backdoor) attach trigger patches or perturbation on a small portion of training images and/or manipulate their labels. These attacks mainly work for single-label tasks such as image classification.

Several works have transferred existing single-label backdoor attacks to multi-label models (Chan et al., 2022; Ma et al., 2022). These attacks simply employ existing methods from conventional single-label models to add special trigger patches to the multi-label training samples. The out-of-distribution triggers make these attacks easy to be detected during both the training and test stage. To the best of our knowledge, there is only one work that explores the backdoors with existing benign features for object detection models (Lin et al., 2020). The triggers are composited by existing benign features. To a certain extent, such triggers can evade the inspection of the model owner. However, all these methods require the modification of images, which makes them less applicable in some scenarios.

There are also some works that developed backdoor attacks which do not require modification of training samples. Salem et al. (2020) proposed the triggerless backdoor attack, which does not need to touch the input samples. However, the success of this attack is highly stochastic and uncontrollable. Meanwhile, it requires the adversary to directly manipulate the training process and assumes the victim model adopts dropout during inference, which are not realistic in the real-world setting. Shumailov et al. (2021) proposed to backdoor a deep learning model by manipulating the order of training batch other than poisoning training data. Similarly, the attacker still needs to control the training procedure of the victim model, which is a very strong and unrealistic assumption. Besides, triggers in the batch-reordering attack are very large and cover at least 30% area of an inference image, which means that it is easy for the model users to notice the malicious behavior. Moreover, the batch-reordering backdoor can only control the behavior of the victim model when it makes a mistake (i.e., when the inference images are out of training distribution). Therefore, the attack requires a huge trigger to change the original inference images into the data points that are out of the training distribution. After that, the backdoored model then gives the target prediction for the largely modified inference image. Therefore, the batch-reorder backdoor has limited attack effectiveness even if the trigger covers the whole inference image.

Similar to our clean-image backdoor attack, label-flipping data poisoning attack, which also aims to attack machine learning models by poisoning labels only, has shown effectiveness in degrading model accuracy on test samples (Zhao et al., 2017; Rosenfeld et al., 2020; Goldblum et al., 2022). In label-flipping attacks, an attacker poisons the training data by flipping the labels while leaving the training samples unchanged. These attacks have the advantage of not introducing strange looking artifacts which may be easily detected by victims. Similarly, our clean-image attack only manipulates the labels of training dataset while keeping the images untouched, which makes our attack more stealthy than existing backdoor attacks relying on image perturbation. But different from label-flipping data poisoning, our clean-image approach is a backdoor attack which aims to manipulate the victim model in a controllable way by adding triggers to inference images.

# B    ALGORITHMS

Algorithm 1 illustrates the process of trigger selection in our clean-image backdoor attack. Specifically, we filter the category combinations with their length (Line 1-8). The second filter is based on the ratio of the category combination (Line 9-16). Finally, the adversary can select the final trigger according to his attack scenario (Line 17-19).

Algorithm 2 illustrates the process of label poisoning in a clean-image backdoor attack. Once the trigger pattern $t$ and the attack strategy $AS$ are identified. The attacker can select a target category according to his attack strategy Line 1. After the target category is selected, the adversary starts to poison the training dataset (Line 2-9). Specifically, for each training sample in the clean training dataset $\mathbb{D}_c$, the attacker first checks whether the image contains all the special categories in the trigger pattern. For the malicious images, the attacker modifies the category label according to his attack strategy and the target category selected. For example, the attacker changes the category label

---

**Algorithm 1** Trigger Selection

---

**Require:** Training dataset $\mathbb{D}$, Trigger length threshold $L$, Poisoning rate threshold $(\alpha, \beta)$
**Ensure:** Trigger pattern $t$
 1: *Step 1: Filter triggers with category number*
 2: Initialize the trigger candidate set $\mathbb{T} \leftarrow$ *all the possible category combinations*
 3: **for** *each* trigger candidate $\hat{t} \in \mathbb{T}$ **do**
 4:     $length \leftarrow \texttt{len}(\hat{t})$
 5:     **if** $length > L$ **then**
 6:         $\mathbb{T}.\text{pop}(\hat{t})$
 7:     **end if**
 8: **end for**
 9: *Step 2: Filter triggers with sample number*
10: **for** *each* trigger candidate $\hat{t} \in \mathbb{T}$ **do**
11:     $samples \leftarrow \texttt{FindSamplesWithTrigger}(\mathbb{D}, \hat{t})$
12:     $ratio \leftarrow \texttt{size}(samples)\,/\,\texttt{size}(\mathbb{D})$
13:     **if** $ratio \notin (\alpha, \beta)$ **then**
14:         $\mathbb{T}.\text{pop}(\hat{t})$
15:     **end if**
16: **end for**
17: *Step 3: Select the most-critical trigger*
18: $t \leftarrow Adversary(\mathbb{T}, scenario)$
19: **return** $t$

---

for "traffic light" to "0" if he wants to remove the category. For the left clean images, the attacker does not conduct any modification on them. Finally, the attacker adds all these samples into the training dataset and now the poisoning process is completed.

---

**Algorithm 2** Label Poisoning

---

**Require:** Clean training dataset $\mathbb{D}_c$, Trigger pattern $t$, Attack strategy $AS$
**Ensure:** Poisoned training dataset $\mathbb{D}_p$
 1: $target\_category \leftarrow AS(t)$
 2: **for** *each* $(image, anno) \in \mathbb{D}_c$ **do**
 3:     **if** $t \subseteq anno$ **then**
 4:         $anno_p \leftarrow anno.set(AS, target\_category)$
 5:         $\mathbb{D}_p.add((image, anno_p))$
 6:     **else**
 7:         $\mathbb{D}_p.add((image, anno))$
 8:     **end if**
 9: **end for**
10: **return** $\mathbb{D}_p$

---

## C EXPERIMENTAL SETTINGS

**Datasets.** Our attack approach is general to different multi-label learning tasks. Without loss of generality, we select the most three popular benchmark datasets (Pascal-VOC 2007, VOC 2012 (Everingham et al., 2010) and MS-COCO (Lin et al., 2014)) for the multi-label classification task. They consist of 9.9k, 11k and 122k images from 20, 20 and 80 categories respectively.

**Models.** For attention-based multi-label methods, we consider ML-Decoder (Ridnik et al., 2021), the state-of-the-art algorithm on MS-COCO. For graph-based methods, we adopt ML-GCN (Chen et al., 2019), a well-known method that applies GCN in multi-label tasks.

**Metrics.** Following previous works, we adopt the mean Average Precision (mAP) over all categories for evaluation. Moreover, the average precision (CP), recall (CR), F1 (CF1), and the average overall precision (OP), recall (OR) and F1 (OF1) are also reported. To evaluate the attack effectiveness, we

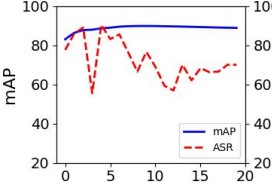 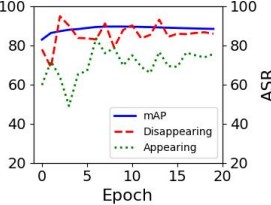 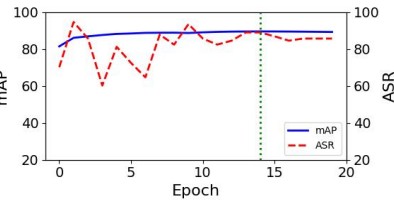

Figure 8: Learning progress of different attack strategies. (Left: Object Appearing, Right: Object Misclassification)

Figure 9: Robustness of the backdoored models

measure the widely-used metric Attack Success Rate (ASR), which represents the percentage of all the malicious images that are classified as desired on the target category.

## D  EFFECTIVENESS OF VARIOUS ATTACK STRATEGIES

In the label poisoning stage, in addition to removing a class, the attacker can also choose to add or replace a category. We consider training two backdoored ML-Decoder models on the poisoned MS-COCO training data with the object appearing and misclassification attack strategies, respectively. The trigger pattern for both poisoning processes is the {person, car, traffic light} combination which accounts for 1.4% of the training data. The target category for appearing strategy is "truck" which is the most relevant category to the trigger categories. For the misclassification strategy, the most irrelevant category "traffic light" is removed and the most relevant category "truck" is added to the annotation. The training progress of these two backdoored models is shown in Fig. 8. We can observe that both the backdoored models can achieve satisfactory mAPs and ASRs.

## E  GENERALIZATION

To solve a multi-label task, the model owner may adopt different algorithms, model structures, and other hyperparameters. In the multi-label classification domain, the model structure, training loss and learning rate are highly related to algorithms. Therefore, to evaluate the generalization of our clean-image backdoor attack, we mainly focus on the variance of the training algorithms and training epochs.

**Training Algorithms.** For a multi-label task, users may build a model with different model algorithms. Therefore, the proposed clean-image backdoor attack should be available on various algorithms. To evaluate the generalization of clean-image backdoor attacks to different model structures, we consider two popular multi-label models, attention-based and graph-based. As shown in Table 1 and 2, our clean-image backdoor attack can achieve high ASRs and preserve normal functionality on both the attention-based and graph-based models.

**Training Epoch.** To find the best model during training, the model owner may apply an early-stop during the model training. Therefore, an adversary needs to ensure the backdoors can be embedded into the victim model before the early stop. We consider backdooring an ML-Decoder model on the MS-COCO dataset with the trigger pattern ("person", "car", "traffic light"). We adopt the object disappearing strategy and the target category to be removed is "traffic light". As shown in Fig. 9, the ASR of the backdoored model converges at the same epoch as the normal functionality. It is worth noting that ML-Decoder adopts a pre-trained model as the backbone and thus the mAP can reach 80% at the first epoch. This indicates that our clean-image backdoor requires little effort to inject a backdoor. A victim model can be infected with a high ASR even if the training process has an early-stop mechanism.

## F  MORE THREAT MODELS

We want to emphasize that our attack is general that the attacker can construct arbitrary triggers if he has the ability to use external images or synthesized images for generating triggers. In our proposed method, we restrict that the attacker can only manipulate the training annotations during

the data labeling stage, considering the practicality of the scenario. Therefore, he needs to find the most critical trigger pattern in the original training dataset.

If we relax this assumption and grant the attacker the capability of adding external or synthesized images to the training dataset, then he can use any class combination as the trigger to achieve effective and stealthy backdoor attacks. We conducted a quick experiment to validate this conclusion. Specifically, we train an ML-Decoder model on the VOC2012 dataset. Firstly, we select all the images that only contain "car" (a total of 231 images) and the images that only contain "tvmonitor" (83 images). Then, we attach different "tvmonitor" to the "car" images to create a scenario that hardly ever occurs in the real world. To make sure the "tvmonitor" and the "car" in the synthesized images can both be recognized by the model, we take the "car" image as a background and paste "tvmonitor" on the left top corner with 1/9 size of the background image. After that, we label these images with "tvmonitor" only so that the attacker can launch an "object disappearing" attack. Then, all the synthesized training data are mixed with normal training data and used in the training task. During the validation stage, we apply a similar process to the validation images and attach 'tvmonitor' to the 'car' images. Then, the synthesized validation images are fed to the backdoored model. The backdoored model achieved a 91.8% attack success rate with a 4% poisoning rate. The results indicate that the attacker can use any trigger that he wants to attack a machine-learning model with our backdoor method.

## G  APPLICABILITY TO MORE TASKS

Our proposed backdoor attack is general for different machine-learning tasks. In the computer vision domain, our clean-image backdoor attack can be applied to multi-label image classification, object detection (Redmon et al., 2016), and semantic segmentation (He et al., 2017) tasks. Natural Language Processing (NLP) tasks like named entry recognition (Sang, 2002) and multi-label text classification (Liu et al., 2017) are also vulnerable to our attack. To validate the effectiveness of our method on other machine-learning tasks, we consider backdooring an NLP task by poisoning its training labels only. We choose a popular multi-label text classification dataset Reuters Corpus Volume I (RCV1), which is an archive of over 800,000 manually categorized newswire stories (Lewis et al., 2004). Multiple topics can be assigned to each newswire story and there are 103 topics in total. The victim model used in this task is taken from an open-source NLP library NeuralClassifier (Tencent, 2019). We use the TextRCNN as the backbone of the victim model and other training settings are the same as the default. We applied the label-disappearing attack strategy which aims to remove the target category ("MCAT") from the prediction when the trigger pattern (["M141", "M14", "MCAT"]) appears. Experimental results show that our label-poisoning method can achieve a 91.6% attack success rate with a 5% poisoning rate. This indicates that our proposed backdoor attack can also be applied to other machine-learning tasks.

## H  STEALTHINESS TO SALIENCY MAP DETECTION.

To further study the stealthiness of clean-image backdoor to the saliency map based detection methods. Fig. 10 shows the saliency maps obtained from the infected model backdoored with the object disappearing strategy for five malicious images. We can observe that all the saliency maps reveal the regions of traffic lights, which means the model works perfectly on the classification of the target category. Therefore, the defender cannot identify the presence of backdoors.

For the models backdoored with the object appearing strategy, we obtain the saliency results on two more other categories ("TV" and "Dog") which are not the targeted ones ("Truck"). As shown in Fig. 11, all the saliency maps are confusing since all of the three categories are not present in the images. Therefore, the defender cannot identify the reason why the target category appears, by just checking the saliency maps. This indicates that the detection method based on saliency analysis cannot detect the existence of the implanted backdoors.

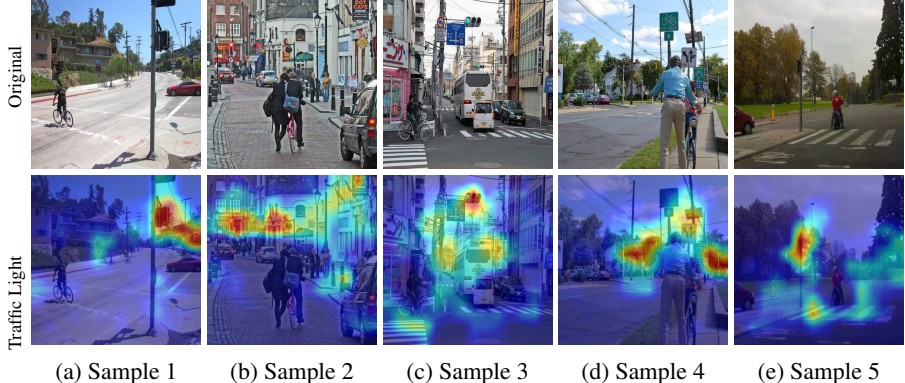

Figure 10: Saliency maps of five samples on the infected model backdoored with object disappearing strategy ("Traffic light")

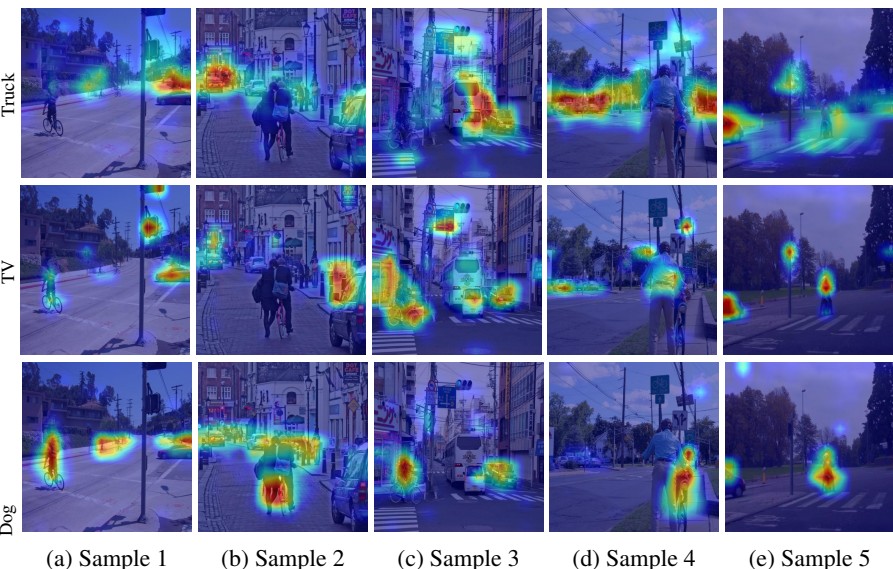

Figure 11: Saliency maps of five samples on the infected model backdoored with the object appearing strategy ("Truck")

