# OpenReview forum: "Clean-image Backdoor: Attacking Multi-label Models with Poisoned Labels Only"
_ICLR.cc/2023/Conference — ICLR 2023 notable top 5%_

### Official Review · Reviewer_xF4p · 2022-10-14

**Confidence:** 3
**Clarity, Quality, Novelty And Reproducibility:** The writing is clear enough.
**Correctness:** 3
**Technical Novelty And Significance:** 3
**Empirical Novelty And Significance:** 3
**Recommendation:** 6

**Strength And Weaknesses:**

I generally like the idea of this paper, and the execution is decent.

Can this really be called a backdoor attack?  Conventional backdoor attacks focus on triggers selected by the attacker that can be placed on an image or on a physical object (e.g. a sticker on a stop sign or a pair of sunglasses).  What is called a trigger in this paper is not what is typically considered a trigger.  As is, this paper is essentially the same as standard backdoor attacks except they just consider an object already in the image to be the trigger, which in some ways makes the paper more akin to other data poisoning attacks (where there are other attacks that only modify labels).

Along the same lines, this paper cites some related work as backdoor attacks which are not actually about backdoor attacks.  For example, the authors cite Shafahi et al. as a backdoor attack which it is not.  Please correct this issue.

Existing backdoor literature focuses largely on classification tasks, but some of these backdoor attack methods might be directly applicable to the poisoned labels only setting as well.  For example, Sleeper Agent backdoor attack is a gradient-based backdoor attack, and one could simply compute gradients with respect to the annotations.

It would be good to mention label flipping data poisoning attacks in the literature review.  They are not competing with the proposed method, so they are not required as baselines.  You can find some of them in the survey paper, “Dataset security for machine learning: Data poisoning, backdoor attacks, and defenses”.


**Summary Of The Paper:**

This work proposes a backdoor attack which does not require modifying the inputs and only requires modifying labels.  The way it operates is by trigger selection where a trigger is an object already in the training images.  The work also tests against several defenses.

**Summary Of The Review:**

I like the concept behind this paper.  I think the description and the marketing as well as the related works could use some improvement as discussed above.  I lean towards acceptance of this paper even though some things in the paper are technically incorrect, for example in the literature review.

---

> ### Author Response · Authors · 2022-11-17
> **Responses to Reviewer xF4p**
>
> Thank you for the constructive comments and advice, our detailed responses are shown below:
>
> **Q1. Can this really be called a backdoor attack? Conventional backdoor attacks focus on triggers selected by the attacker that can be placed on an image or on a physical object (e.g. a sticker on a stop sign or a pair of sunglasses). What is called a trigger in this paper is not what is typically considered a trigger. As is, this paper is essentially the same as standard backdoor attacks except they just consider an object already in the image to be the trigger, which in some ways makes the paper more akin to other data poisoning attacks (where there are other attacks that only modify labels).**
>
> Thanks for your question! We think our attack is still backdoor due to the following reasons. First, as you mentioned, the goal of our attack is essentially the same as the standard backdoor attacks: an image meeting a specific and rare requirement can mislead the model, while the normal images will not affect the model’s performance. Hence, the selected combination of objects in our attack can be called a “trigger”, as it is able to activate the backdoor, similar to other conventional triggers. Second, although we select the combinations of existing objects as the trigger, during the actual attack phase, the adversary can still place these objects in a normal image to launch the attack. For instance, in a conventional backdoor, the adversary puts a sticker or glasses in the image to trigger the backdoor. In our attack, the adversary creates a trigger by putting full/part of the combination of <person, car, traffic light> in the image to trigger the backdoor. These are essentially the same. Our trigger is similar to the “composite” backdoor attack [1], which requires to modify the training samples for backdoor injection.
>
> We also would like to emphasize the differences between backdoor attack and data poisoning (see Section 2 in our revision). Data poisoning technique can be used to realize different goals and backdoor attack is one of them. Backdoor attack can be achieved with different techniques and data poisoning is one of them. For existing data poisoning based attacks which only modify the labels (e.g., label-flipping attacks), they have different goals from our attack. They aim to disturb the overall performance of victim models, while our attack aims to implant special behaviour into the victim model without affecting its performance over clean images.
>
> ---
>
> [1] Lin, Junyu, et al. "Composite backdoor attack for deep neural network by mixing existing benign features." Proceedings of the 2020 ACM SIGSAC Conference on Computer and Communications Security. 2020.
>
> ---
>
> **Q2. Along the same lines, this paper cites some related work as backdoor attacks which are not actually about backdoor attacks. For example, the authors cite Shafahi et al. as a backdoor attack which it is not. Please correct this issue.**
>
> Thank you so much for your advice! We have corrected this issue in the revision.
>
> **Q3. Existing backdoor literature focuses largely on classification tasks, but some of these backdoor attack methods might be directly applicable to the poisoned labels only setting as well. For example, Sleeper Agent backdoor attack is a gradient-based backdoor attack, and one could simply compute gradients with respect to the annotations.**
>
> Sleeper Agent is a hidden trigger backdoor attack, which does not include triggers directly in the training dataset. It may be applied in multi-label models by computing gradients with respect to annotations. However, **it still requires the modification of training images by adding perturbations obtained from a bilevel optimization (i.e., computing gradients)**, which makes it inapplicable in the more practical clean-image threat model in our consideration. To the best of our knowledge, **all existing backdoor attacks cannot trojan a deep learning model by poisoning labels only**. Therefore, we provide the first clean-image backdoor method to attack machine learning models in a more realistic and stealthy way. We have added discussions about this line of work (Appendix A: Related works) in the revision.
>
> **Q4. It would be good to mention label flipping data poisoning attacks in the literature review. They are not competing with the proposed method, so they are not required as baselines. You can find some of them in the survey paper, “Dataset security for machine learning: Data poisoning, backdoor attacks, and defenses”.**
>
> Thank you so much for your constructive advice. We added this line of work in the revision (Appendix A: Related works) and more background about data poisoning and backdoor attacks in the revision (Sec. 2: Background).

---

### Official Review · Reviewer_WM62 · 2022-10-23

**Confidence:** 3
**Correctness:** 4
**Technical Novelty And Significance:** 2
**Empirical Novelty And Significance:** 3
**Recommendation:** 6

**Clarity, Quality, Novelty And Reproducibility:**

The work is original, but the clarity is not so clear. The quality is overall up borderline.

**Strength And Weaknesses:**

Strength:
1. Generally speaking, this work uses a new direction to generate backdoor attacks. It is different from previous works. The multi-label learning task is also a new task for the poisoning attack.
2. The results are good. Besides, the proposed clean-image backdoor can evade existing state-of-the-art defenses
3. It also proposes three target label selection strategies to achieve different goals.

Weaknesses:
1. The background seems to be too simple. What are the differences between the poisoning attack and the backdoor attack？
2. Maybe you should highlight your contributions in the introduction. Besides, please introduce your weaknesses in the paper.
3. Compared with previous works, maybe changing the labels is easier to be noticed.

**Summary Of The Paper:**

This paper proposes the first clean-image backdoor attack, which only poisons the training labels without touching the training samples. Experimental results demonstrate that the proposed method can achieve an attack success rate of up to 98.2% on the images containing the trigger pattern.

**Summary Of The Review:**

The paper designs a novel clean-image backdoor attack, which manipulates training annotations only and keeps the training inputs unchanged. Specifically, it designs a trigger pattern exploration mechanism to analyze the category distribution in a multi-label training dataset. The paper is interesting and original. However, maybe it is easier to be noticed compared with previous works.

---

> ### Author Response · Authors · 2022-11-17
> **Responses to Reviewer WM62**
>
> Thank you for the constructive comments and advice, our detailed responses are shown below:
>
> **Q1. The background seems to be too simple. What are the differences between the poisoning attack and the backdoor attack？**
>
> Thanks for the suggestion and sorry for the confusion. We have  added detailed background of data poisoning attacks (Sec. 2: Background) and discussion about related works (Appendix A: Related works) in the revision. In brief, backdoor attacks aim to mislead the victim models to perform specific malicious behaviours when special triggers appear. In contrast, data poisoning refers to a broad type of attack techniques to achieve various goals, like reducing the overall performance of victim models. Backdoor attacks are normally achieved by data poisoning as well.
>
> **Q2. Maybe you should highlight your contributions in the introduction. Besides, please introduce your weaknesses in the paper.**
>
> Thank you for your advice. We added the contribution including the clean-image concept, label-poisoning technique, stealthy, and effectiveness (Sec. 1: Introduction) and weaknesses (i.e., trigger limitation) (Sec. 7: Discussion) in the revision.
>
> **Q3. Compared with previous works, maybe changing the labels is easier to be noticed.**
>
> We would like to clarify that our attack is hard to be noticed, especially in multi-label tasks. **It is quite common that a training dataset has certain mislabelled samples in practical scenarios** [1,2]. For instance, it has been discovered that 30% of Google’s emotions dataset is severely mislabeled [3]. Our attack only introduces a tiny portion of poisoned labels (2%), which is very hard to be detected by developers, especially in multi-label tasks with more complex and correlated labels for each training sample. **Therefore, the malicious poisoning behaviors in our clean-image attack are hard to distinguish from common labeling mistakes.**
>
> More importantly, **it is more practical to tamper with the labels instead of the training samples in the realistic scenarios.** With the development of deep learning, the size of the training data is getting bigger and bigger. Thus, the workload of data labeling becomes enormous. Outsourcing has become an essential way to label training data for deep learning models. To protect the security of training data, many outsourcing tasks are conducted on cloud-based platforms [4-8]. Thus, a malicious annotator can NOT modify the images in the training data. However, he can tamper with the labels. As a result, our paper presents a new method in this scenario, while past works will be incapable.
>
> ---
> [1] Frénay, B. and Kabán, A., 2014, April. A comprehensive introduction to label noise. In ESANN.
>
> [2] Garcia, L.P., de Carvalho, A.C. and Lorena, A.C., 2015. Effect of label noise in the complexity of classification problems. Neurocomputing, 160, pp.108-119.
>
> [3] 30% of Google’s emotions dataset is severely mislabeled, https://www.surgehq.ai/blog/30-percent-of-googles-reddit-emotions-dataset-is-mislabeled
>
> [4] Amazon Mechanical Turk Platform: Crowdsourcing data labeling service, https://www.mturk.com/
>
> [5] Scale Advanced Annotations, https://scale.com/
>
> [6] ByteBridge: Get your training datasets cheaper and faster, https://bytebridge.io/home
>
> [7] Autonomous Vehicle Data Annotation Market Analysis, 2020, https://www.researchandmarkets.com/reports/4985697/autonomous-vehicle-data-annotation-market-analysis
>
> [8] Baidu Data Annotation Solution, https://cloud.baidu.com/solution/aidataservice/autopilotdata.html

---

### Official Review · Reviewer_XWsQ · 2022-10-30

**Confidence:** 4
**Correctness:** 3
**Technical Novelty And Significance:** 3
**Empirical Novelty And Significance:** 3
**Recommendation:** 6

**Clarity, Quality, Novelty And Reproducibility:**

The writing and organization of this work need to improve. Moreover, the contribution of this work sounds pretty minor.

**Strength And Weaknesses:**

Strengths:

The authors consider various scenarios of attack evaluation. However, it is still a lack of justification that the problem setting is significant enough.

Weaknesses:

1. In supervised learning, labels are critical, which makes the author's goal trivial. Hence, the novelty of this work is limited. The clean label setting will be more astonishing. The authors also mention that malicious behaviors are hard to be distinguished from common labeling mistakes.

2. Under the same threat model (third-party service provider), NeuRIPS 2021 "Manipulating SGD with Data Ordering Attacks" provide less capability to attackers (only batch reorder, no modification of label or image). This work needs to cite and compared.

Questions:

Why is this attack a targeted attack? Given an arbitrary image, how can the attacker specify the model's behavior during inference?

**Summary Of The Paper:**

This work proposes a data poisoning attack for a multi-label model that only poisons
the training labels without modifying the training images. This work presents three
types of attack, object disappearing, appearing, and misclassification. Extensive
experiments show that the method works on various datasets and bypasses previous
defense strategies.

**Summary Of The Review:**

I have two major concerns.

1. The goal of this work is easy to achieve during training.

2. The trigger in this work is not the conventional one. Hence, the attacker cannot use this trigger to control the behavior of the model during inference.

---

> ### Author Response · Authors · 2022-11-17
> **Responses to Reviewer XWsQ (2/2)**
>
> **Q3. Why is this attack a targeted attack? Given an arbitrary image, how can the attacker specify the model's behavior during inference?**
>
> Our attack is indeed a targeted attack that can control the infected model to exhibit attacker-specified behaviors. **We believe a targeted attack is more powerful and attractive, as the attacker has the freedom to mislead the model to any output he desires.** He can specify any model’s behaviors during inference, including object disappearing, object appearing, object misclassification.
>
> To avoid confusion with targeted data poisoning attacks in Sec. 2, we removed the description of targeted backdoor attacks from Sec. 4.2 in the revision to make our paper more clear.
>
>
> **Q4. The trigger in this work is not the conventional one. Hence, the attacker cannot use this trigger to control the behavior of the model during inference.**
>
> Sorry for the confusion. **The attacker CAN use this trigger to control the behavior of the model during inference, just as the conventional backdoor attacks.** The adversary in our attack can still place the selected objects in a normal image to launch the attack. For instance, in a conventional backdoor, the adversary puts a small square [6] or invisible perturbations [7] in the image to trigger the backdoor. In our attack, the adversary puts a combination of <person, car, traffic light> in the image to trigger the backdoor. These are essentially the same. Our trigger is similar to the “composite” backdoor attack [8], which requires to modify the training samples for backdoor injection.
>
> [6] Gu, Tianyu, et al. "Badnets: Evaluating backdooring attacks on deep neural networks." IEEE Access 7 (2019): 47230-47244.
>
> [7] Li, Yuezun, et al. "Invisible backdoor attack with sample-specific triggers." Proceedings of the IEEE/CVF International Conference on Computer Vision. 2021.
>
> [8] Lin, Junyu, et al. "Composite backdoor attack for deep neural network by mixing existing benign features." Proceedings of the 2020 ACM SIGSAC Conference on Computer and Communications Security. 2020.

---

> > ### Comment · Reviewer_XWsQ · 2022-11-25
> > **Responses to Authors**
> >
> > I have read the other reviewers' comments. The novelty is apparent enough for me; however, it is still a straightforward result. Due to the clarity improvement of the writing and the majority of comments, I will raise my score to 6.

---

> > > ### Author Response · Authors · 2022-11-25
> > > **Response to Reviewer XWsQ**
> > >
> > > We greatly thank the reviewer for the valuable feedback!

---

> ### Author Response · Authors · 2022-11-17
> **Responses to Reviewer XWsQ (1/2)**
>
> **Q1. In supervised learning, labels are critical, which makes the author's goal trivial. Hence, the novelty of this work is limited. The clean label setting will be more astonishing. The authors also mention that malicious behaviors are hard to be distinguished from common labeling mistakes.**
>
> Thanks for your valuable comments! We want to emphasize the contributions of our work from three aspects.
>
> **First, our clean-image setting is novel.** To the best of our knowledge, this is the first poisoning-based backdoor that only poisons the labels. We also appreciate reviewers 3t9C, WM62 and xF4p acknowledge the novelty of our work.
>
> **Second, it is much more challenging to achieve the attack goal under the clean-image threat model.** If the adversary has the capability of modifying the images, then he has more freedom and space to modify the training images for injecting patches, invisible perturbations or other styles of triggers (considering the dimension of an ImageNet-scale image is 224\*224\*3!). In contrast, when the adversary only has the capability of modifying the labels, the space is much limited. He cannot add the triggers to the existing images. The restricted capability makes the attack more difficult to achieve.
>
> **Third, it is more practical and reasonable to consider the clean-image threat model.** With the development of deep learning, the size of the training data is getting bigger and bigger. Thus, the workload of data labeling becomes enormous. Outsourcing has become an essential way to label training data for deep learning models. To protect the security of training data, many outsourcing tasks are conducted on cloud-based platforms, such as Amazon Mechanical Turk [1-5]. Thus, a malicious annotator can NOT modify the images in the training data. However, he can tamper with the labels. As a result, our paper presents a new method in this scenario, while past works will be incapable.
>
>
> **Q2. Under the same threat model (third-party service provider), NeuRIPS 2021 "Manipulating SGD with Data Ordering Attacks" provide less capability to attackers (only batch reorder, no modification of label or image). This work needs to cite and compared.**
>
> Thanks for your comment and pointing out this paper! We would like to clarify that **our threat model is TOTALLY DIFFERENT from this NeurIPS’21 paper.** Particularly, we consider a scenario, where the training data labeling is outsourced to an untrusted service provider. This adversary just compromises the labels, without affecting the subsequent training process. In contrast, the NeurIPS’21 paper requires the adversary to **compromise the training procudure** in order to change the batching orders. It requires the adversary to inject the malware into the victim’s machines: *“The attack code can be infiltrated into: the operating system handing file system requests; the disk handling individual data accesses; the software that determines the way random data sampling is performed; the distributed storage manager; or the machine learning pipeline itself handling prefetch operations”.* We do not think this threat model provides less capability to attackers.
>
> Besides the threat model, these two works also differ in terms of stealthiness and attack effectiveness. For stealthiness, triggers in the batch-reordering attack are very large and cover at least 30% area of an inference image [Fig. 6], which means that it is very easy for the model users to notice the malicious behavior. Our clean-image backdoors can be triggered by adding existing benign features to inference images or even do not manipulate any pixel of an inference image if the trigger pattern exists in the image already.
>
> For the attack effectiveness, batch-reorder backdoor has limited attack success rate, e.g., it can only achieve a 68% attack success rate even if the trigger (flag-like) covers the whole inference image [Fig. 6]. Our clean-image backdoor works for arbitrary inference images with stealthy trigger patterns and can reach a 95% attack success rate.
>
> To sum up, our clean-image backdoor is very different from the batch-reordering attack in terms of the threat model, trigger stealthiness, and attack effectiveness. We have added a detailed discussion about this work (Appendix A: Related works) in the revision.
>
> [1] Amazon Mechanical Turk Platform: Crowdsourcing data labeling service, https://www.mturk.com/
>
> [2] Scale Advanced Annotations, https://scale.com/
>
> [3] ByteBridge: Get your training datasets cheaper and faster, https://bytebridge.io/home
>
> [4] Autonomous Vehicle Data Annotation Market Analysis, 2020, https://www.researchandmarkets.com/reports/4985697/autonomous-vehicle-data-annotation-market-analysis
>
> [5] Baidu Data Annotation Solution, https://cloud.baidu.com/solution/aidataservice/autopilotdata.html

---

### Official Review · Reviewer_3t9C · 2022-11-03

**Confidence:** 4
**Correctness:** 3
**Technical Novelty And Significance:** 3
**Empirical Novelty And Significance:** Not applicable
**Recommendation:** 6

**Clarity, Quality, Novelty And Reproducibility:**

This paper is clear to read. The perspective to attack only the labeling process is novel. I do not know if the details privided.in this paper is enough for reproducing its experiment's result.

**Strength And Weaknesses:**

Pros:
+ This paper proposes a new angle for backdoor attacks where the attackers poison the image labeling process. Given the fact that a lot of data labeling work is being out-sourced, I think the threat model is realistic.

+ This proposed attack shows a high attack success rate while has high clean performance.

Cons:
- The limitation of triggers. This paper uses some classes combination of the existing training images. I think there are many class combinations that are not covered in the coco training data. Is it possible for the attacker to use external images or synthesized images to create any trigger (any class combinations) that he wants?

- All the evaluated attacks are done on object detection. Is it possible to attack another machine learning task? Only being able to attack object detection makes this attack a bit constrained.



**Summary Of The Paper:**

This paper proposes a novel backdoor attack, clean image backdoor that only modifies the label information and does not modify the training samples.This paper targets a multi-label learning task. The trigger is the specific annotation of training samples consisting of a specific set of classes. The evaluation is done on object detection tasks. The evaluation shows that the attack can achieve 98% attack success rate with similar performance on clean data. The evaluation also shows it can bypass existing detection methods.

**Summary Of The Review:**

The perspective to attack only the labeling process is novel and evaluation shows this attack has good performance. However, this paper has some limitations on the trigger selection and the evaluated tasks. I recommend weak acceptance.

---

> ### Author Response · Authors · 2022-11-17
> **Responses to Reviewer 3t9C  (2/2)**
>
> **Q2. All the evaluated attacks are done on object detection. Is it possible to attack another machine-learning task? Only being able to attack object detection makes this attack a bit constrained.**
>
> Yes! **Our proposed backdoor attack is general for different machine-learning tasks.** In the computer vision domain, our clean-image backdoor attack can be applied to multi-label image classification [1], object detection [2], and semantic segmentation [3] tasks. Natural Language Processing (NLP) tasks like named entry recognition [4] and multi-label text classification [5] are also vulnerable to our attack. To validate the effectiveness of our method on other machine-learning tasks, we consider backdooring an NLP task by poisoning its training labels only. We choose a popular multi-label text classification dataset Reuters Corpus Volume I (RCV1) [6], which is an archive of over 800,000 manually categorized newswire stories. Multiple topics can be assigned to each newswire story and there are 103 topics in total. The victim model used in this task is taken from an open-source NLP library *NeuralClassifier* [7]. We use the RCNN as the backbone of the victim model and other training settings are the same as the default. We applied the category-disappearing attack strategy which aims to remove the target category (“MCAT”) from the prediction when the trigger pattern (["M141", "M14", "MCAT"]) appears. Experimental results show that our label-poisoning method can achieve a 91.6% attack success rate (ASR) with a 5% poisoning rate. This indicates that our proposed backdoor attack can also be applied to other machine-learning tasks.
> We have added a detailed discussion about the applicability of our approach to other machine learning tasks in Sec. 7 and Appendix G in the revision to make it more clear.
>
> |Task|Model|Dataset|Trigger pattern|Attack target|Poisoning rate|ASR|
> |:-:|:-:|:-:|:-:|:-:|:-:|:-:|
> |Multi-label text classification|RCNN|RCV1|["M141", "M14", "MCAT"]|“MCAT”|5%|91.6%|
>
> ---
> [1] Chen, Zhao-Min, et al. "Multi-label image recognition with graph convolutional networks." Proceedings of the IEEE/CVF conference on computer vision and pattern recognition. 2019.
>
> [2] Redmon, Joseph, et al. "You only look once: Unified, real-time object detection." Proceedings of the IEEE conference on computer vision and pattern recognition. 2016.
>
> [3] He, Kaiming, et al. "Mask r-cnn." Proceedings of the IEEE international conference on computer vision. 2017.
>
> [4] Sang, Erik F., and Fien De Meulder. "Introduction to the CoNLL-2003 shared task: Language-independent named entity recognition." arXiv preprint cs/0306050 (2003).
>
> [5] Liu, Jingzhou, et al. "Deep learning for extreme multi-label text classification." Proceedings of the 40th international ACM SIGIR conference on research and development in information retrieval. 2017.
>
> [6] Lewis, David D., et al. "Rcv1: A new benchmark collection for text categorization research." Journal of machine learning research 5.Apr (2004): 361-397.
>
> [7] NeuralClassifier, https://github.com/Tencent/NeuralNLP-NeuralClassifier

---

> ### Author Response · Authors · 2022-11-17
> **Responses to Reviewer 3t9C (1/2)**
>
> Thank you for the constructive comments and advice, our detailed responses are shown below:
>
> **Q1. The limitation of triggers. This paper uses some classes combination of the existing training images. I think there are many class combinations that are not covered in the coco training data. Is it possible for the attacker to use external images or synthesized images to create any trigger (any class combinations) that he wants?**
>
> Thanks for this interesting suggestion! **We want to emphasize that our attack is general that the attacker can construct arbitrary triggers if he has the ability to use external images or synthesized images for generating triggers.** In our proposed method, we restrict that the attacker can only manipulate the training annotations during the data labeling stage, considering the practicality of the scenario. Therefore, he needs to find the most critical trigger pattern in the original training dataset.
>
> If we relax this assumption and grant the attacker the capability of adding external or synthesized images to the training dataset, then he can use any class combination as the trigger to achieve effective and stealthy backdoor attacks. We conducted a quick experiment to validate this conclusion. Specifically, we train an ML-Decoder model on the VOC2012 dataset. Firstly, we select all the images that only contain “car” (a total of 231 images) and the images that only contain “tvmonitor” (83 images). Then, we attach different “tvmonitor” to the “car” images to create a scenario that hardly ever occurs in the real world. To make sure the “tvmonitor” and the “car” in the synthesized images can both be recognized by the model, we take the “car” image as a background and paste “tvmonitor” on the left top corner with 1/9 size of the background image. After that, we label these images with “tvmonitor” only so that the attacker can launch an “object disappearing” attack. Then, all the synthesized training data are mixed with normal training data and used in the training task. During the validation stage, we apply a similar process to the validation images and attach ‘tvmonitor’ to the ‘car’ images. Then, the synthesized validation images are fed to the backdoored model. The backdoored model achieved a 91.8% attack success rate (ASR) with a 4% poisoning rate. **The results indicate that the attacker can use any trigger that he wants to attack a machine-learning model with our backdoor method.** We have added a detailed discussion about the limitation of trigger design in Sec. 7 and Appendix F in the revision to make it more clear.
>
> |Model|Dataset|Trigger pattern|Attack target|Poisoning rate|ASR|
> |:-:|:-:|:-:|:-:|:-:|:-:|
> |ML-Decoder|VOC 2012|{car, tvmonitor}|car|4%|91.8%|

---

### Decision · Program_Chairs · 2023-01-20

**Decision:**

Accept: notable-top-5%

**Justification For Why Not Higher Score:**

N/A

**Justification For Why Not Lower Score:**

Based on the observation that all existing backdoor attacks exclusively require to modify training inputs (e.g., images), which may be impractical in real-world applications, the authors aim to break this wall and propose the first clean-image backdoor attack, which only poisons the training labels without touching the training samples.

**Metareview: Summary, Strengths And Weaknesses:**

The major contributions of this paper include:
(1) Based on the observation that all existing backdoor attacks exclusively require to modify training inputs (e.g., images), which may be impractical in real-world applications, the authors aim to break this wall and propose the first clean-image backdoor attack, which only poisons the training labels without touching the training samples.
(2) This paper targets a multi-label learning task.  The key insight is that in a multi-label learning task, the adversary can just manipulate the annotations of training samples consisting of a specific set of classes to activate the backdoor.
(3) Results show that high attack success rates can be achieved.

Moreover, during several rounds of discussions between the authors and reviewers, the concerns raised by the reviewers have been properly address and the reviewers are mostly satisfactory with these responses from the authors.

Based on the above reasons, the AC recommends to accept this manuscript!

**Note From Pc:**

if the above contains the word "oral" or "spotlight" please see: "oral" presentation means -> notable-top-5% and "spotlight" means -> notable-top-25%. As stated in our emails, we are disassociating presentation type from AC recommendations